# Differences in Anxiety, Insomnia, and Trauma Symptoms in Wildfire Survivors from Australia, Canada, and the United States of America

**DOI:** 10.3390/ijerph21010038

**Published:** 2023-12-27

**Authors:** Fadia Isaac, Samia R. Toukhsati, Britt Klein, Mirella Di Benedetto, Gerard A. Kennedy

**Affiliations:** 1Institute of Health and Wellbeing, Federation University, Mt Helen, VIC 3350, Australiag.kennedy@federation.edu.au (G.A.K.); 2Health Innovation and Transformation Centre, Federation University, Mt Helen, VIC 3350, Australia; b.klein@federation.edu.au; 3Biopsychosocial & eHealth Research & Innovation (BeRI) Hub, Federation University, Mt Helen, VIC 3350, Australia; 4Australian Centre for Heart Health, North Melbourne, VIC 3051, Australia; mirelladb25@gmail.com; 5School of Health and Biomedical Sciences, RMIT University, Melbourne, VIC 3083, Australia; 6Institute for Breathing and Sleep, Austin Health, Heidelberg, Melbourne, VIC 3084, Australia

**Keywords:** depression, anxiety, PTSD, nightmares, insomnia, sleep quality, wildfires, survivors, USA, Canada, Australia

## Abstract

Many survivors of wildfires report elevated levels of psychological distress following the trauma of wildfires. However, there is only limited research on the effects of wildfires on mental health. This study examined differences in anxiety, depression, insomnia, sleep quality, nightmares, and post-traumatic stress disorder (PTSD) symptoms following wildfires in Australia, Canada, and the United States of America (USA). One hundred and twenty-six participants from Australia, Canada, and the USA completed an online survey. The sample included 102 (81%) women, 23 (18.3%) men, and one non-binary (0.8%) individual. Participants were aged between 20 and 92 years (*M* age = 52 years, *SD* = 14.4). They completed a demographic questionnaire, the Disturbing Dream and Nightmare Severity Index (DDNSI), Generalized Anxiety Disorder Questionnaire (GAD-7), the Insomnia Severity Index (ISI), Patient Health Questionnaire (PHQ-9), the Pittsburgh Sleep Quality Index (PSQI), and PTSD Checklist (PCL-5). Results showed that participants from the USA scored significantly higher on the GAD-7 (*p* = 0.009), ISI (*p* = 0.003), and PCL-5 (*p* = 0.021) than participants from Australia and Canada. The current findings suggest a need for more international collaboration to reduce the severity of mental health conditions in Australia, Canada, and the USA.

## 1. Introduction

Wildfires are vital events for many ecosystems in preserving species that respond to fires, stimulating seed germination and growth of native vegetation, helping to eliminate competition from invasive weeds, and eradicating diseases and insects that cause harm to older plants and vegetation [1,2,3]. However, when wildfires spread rapidly with great intensity and force, they annihilate forests, wildlife, and entire communities. This decade has witnessed unparalleled numbers of wildfires affecting the globe including; the Arctic, the United States of America (USA), Canada, parts of Europe, and Australia [4,5,6].

In Australia, the 2019–2020 Black Summer fires resulted in the burning of more than 24 million hectares of land, destroyed 3000 homes, and killed 33 people [7,8]. Similarly, in 2018, British Columbia/Canada was hit by the worst wave of wildfires in the region’s recorded history, leading to the destruction of 1.35 million hectares of land, destroying 2211 properties, and USD 615 million was spent to fight the fires [5]. Furthermore, The August Complex Fires in the USA in 2020 were labeled the largest wildfires that the state had ever witnessed. It led to the burning of 1.6 million hectares of land, destroyed 8200 buildings, killed 31 people, and displaced tens of thousands of people for several months following the fires [9,10,11]. The three countries suffered major financial and biodiversity losses. The consequences of wildfires have major negative effects on the mental and physical health of survivors by disrupting social networks and causing financial losses and hardship that may persist for decades [6].

Numerous studies suggest that the magnitude of suffering for survivors is associated with geographic proximity to wildfires and the extent and number of losses incurred during the fires [12,13,14,15]. The level of suffering is not only limited to financial losses but also to the negative impact that wildfires impose on the physical and mental wellbeing of survivors. Following the trauma of wildfires, many wildfire survivors report elevated levels of anxiety, depression, stress, sleep difficulties, and post-traumatic stress disorder (PTSD) symptoms [12,13]. In a comprehensive review of 63 studies that examined the impact of wildfires on mental health, To et al. [16] found that the rates of PTSD ranged between 29% and 60% at 3 months, 12.8% to 26% at 6 months, and 15.6% to 7.6% at 3–10 years following the trauma of wildfires; high rates of depression were also reported following the fires, with percentages ranging between 25.5% and 33% at 3 months, 10.4% and 17.1% at 6 months, and approximately 10% at 10 years following the disaster. Similarly, anxiety was also reported following the fires, with approximately 17.4% to 27.0% of survivors reporting symptoms at 3 months, 19.8% at 6 months, and 4.4% to 7.5% at 10 years post-wildfires [16]. Symptoms of insomnia and nightmares were also found to be some of the most prevalent mental health conditions reported by survivors following the trauma of wildfires. For example, the incidence of insomnia was found to range between 28.5% and 77.9%, and the incidence of nightmares ranged between 33.3% and 49.2% following the disaster [13,14,15,17,18].

The trauma experienced by survivors in the period following the fires is not the sole contributor to the high rates and the severity of the mental health conditions reported. Studies show that a constellation of other external factors contributed to and/or intensified the impact of the trauma of wildfire by increasing stress levels in affected individuals. Some of those factors included younger age, being a female, low education levels, loss of a job, job stress and job relocation, limited social support, low socioeconomic status, prior mental health history, and childhood trauma. Experiencing one or more of those factors can lead to higher rates and more severe presentation of conditions such as PTSD, depression, and anxiety [16,18,19,20,21]. Recency of wildfires also seemed to be a major contributor in dictating the rates and severity of mental health conditions reported by individuals who experienced wildfires not just within the first 12 months but also in the years following the trauma of wildfires [16,18].

Most findings about the effect of wildfires on mental health are mainly drawn from survivors in countries that are most severely affected by wildfires, including Australia, Canada, and the USA [16]. However, comparing the severity of mental health conditions after wildfires between the three countries is poorly researched and understood. One reason for this is that researching mental health in wildfire survivors can be challenging due to ongoing symptoms of trauma that are common, with many survivors wishing to avoid re-visiting traumatizing events. Therefore, it is not surprising that cross-cultural research is limited in exploring how people in different countries with different social structures may be affected by wildfires.

Differences in rates and severity of mental health conditions between different countries may be expected due to not only differences in policies applied in each country but also the level of preparedness implemented in each country in relation to wildfires. The level of preparedness for fires can act as a buffer against the long-term and largely ignored negative consequences on mental health in vulnerable communities [19,22].

Thus, the main objective of the current study was to compare mental health outcomes following wildfires in Australia, Canada, and the USA. Specifically, the aim was to examine patterns of severity and differences in anxiety, depression, insomnia, sleep quality, nightmares, and PTSD symptoms. Comparing health data across countries can support decision making and policy planning for those at risk of experiencing wildfires [23]. Furthermore, a comparison of mental health conditions between Australia, Canada, and the USA may provide useful information to inform the international community about the likelihood of mental health outcomes following wildfires and other natural disasters.

## 2. Method

### 2.1. Participants

The participants were 126 wildfire survivors from Australia, Canada, and the USA. Twenty-three males (18.3%), 102 (81%) females, and one nonbinary (0.8%) individual took part in this study. Forty-four (34.9%) participants from Australia, 27 (21.4%) from Canada, and 55 (43.7%) from the USA completed an online survey. Participants ages ranged between 20 and 92 years (*M* age = 52 years, *SD* = 14.4).

### 2.2. Measures

Demographic questions: Demographic information was collected from participants such as age, gender, country of residence, education level (no schooling, primary, secondary, certificate or diploma, bachelor’s degree, or postgraduate degree), employment history (student, employed, unemployed, looking for work, or retired), income (six categories were adapted from the Australian Bureau of Statistics ranging between $AUD 0 and 156,000 or more per year, converted to $USD for each country during the analysis), and recency of wildfires (participants were asked to provide the dates of the wildfires they had experienced in the last 10 years, which were divided into two categories: wildfires experienced less than 12 months ago and wildfires experienced more than 12 months ago) [16,18].

Disturbing Dream and Nightmare Severity Index (DDNSI): The scale consists of five self-reported items assessing the frequency and severity of disturbing dreams and nightmares [24]. The DDNSI assesses the number of nights with nightmares per week (0–7 nights) and number of nightmares per week (0–14 nightmares). The DDNSI also assesses the intensity and severity of nightmares on a Likert-type scale (0 = no problems to 6 = extremely severe) and nightmare awakenings (0 = never or rarely to 4 = always). Scores range 0–37, with scores greater than 10 reflecting the presence of a nightmare disorder [24]. A previous study showed that the DDNSI had a Cronbach’s alpha of α = 0.93 [25].

Generalized Anxiety Disorder Questionnaire (GAD-7): The GAD-7 consists of seven self-reported items that assess worry and anxiety symptoms. Items are rated on a 4-point Likert scale from 0 = not at all to 3 = nearly every day [26]. Scores range from 0 to 21, with higher scores indicating more severe symptoms of anxiety. The scores fall into one of four ranges, with 0–4 indicating minimal anxiety, 5–9 reflecting mild anxiety, 10–14 representing moderate anxiety, and scores from 15–21 reflecting severe anxiety symptoms. The GAD-7 has been found to be a valid screening tool for anxiety in primary care settings and for assessing severity in clinical practice and research [26]. A cut-off score of 10 has been identified as the optimal point for sensitivity of 89% and specificity of 82% [26]. Cronbach’s alpha was found to be α = 0.95 for the GAD-7 in the current sample.

The Insomnia Severity Index Scale (ISI): The ISI is a short self-report questionnaire measuring symptoms and severity of insomnia [27]. The ISI is composed of seven items assessing problems with sleep onset, sleep maintenance, early morning awakening, interference of sleep problems with daily functioning, concern about sleep problems, and satisfaction with sleep patterns over the last month. The severity of each item is rated on a scale from 0 to 4. Total score ranges from 0 to 28, whereby higher scores suggest more severe symptoms. The ISI consists of four categories: 0–7 = no clinical insomnia, 8–14 = subthreshold insomnia, 15–21 = clinical insomnia/moderate severity, and 22–28 = clinical insomnia/severe [28]. A cut-off score of 14 provides 82.4% sensitivity and 82.1% specificity for detecting clinical insomnia [29]. In the present sample, Cronbach’s alpha was α = 0.92. 

The Patient Health Questionnaire (PHQ-9): Nine self-reported items are used in this scale to measure symptoms of depression [30]. Items are rated on a 4-point Likert scale (0 = not at all to 3 = nearly every day). Total scores range from 0 to 27. Scores higher than 10 indicate the presence of depressive disorder [30]. Kroenke and colleagues [30] suggest the following levels of severity: scores ranging between 1 and 4 = minimal; 5 to 9 = mild; 10 to 14 = moderate; 15 to 19 = moderately severe; and 20 to 27 = severe. Cronbach’s alpha for the PHQ-9 in the current study was α = 0.91.

Pittsburgh Sleep Quality Index (PSQI): The scale consists of 19 self-reported items with an additional five questions rated by a bed partner [31]. The PSQI is scored on a Likert-type scale ranging from 0 to 3. It assesses seven components of sleep quality in the past month, including: subjective sleep quality, sleep latency, sleep duration, sleep efficiency, sleep disturbances, use of sleep medication, and impairment in daytime functioning. A global sleep quality score ranges between 0 and 21, and it is obtained by summing the seven component scores. Higher scores indicate poorer sleep quality. A global PSQI score greater than 5 indicates a diagnostic specificity of 84.4% and a sensitivity of 98.7% in distinguishing between “good” and “poor” sleepers [32]. In the current sample, Cronbach’s alpha for the PSQI was α = 0.81.

PTSD Checklist for DSM-5 Scale (PCL-5, Civilian Version): Providing a provisional diagnosis of PTSD, the PCL-5 consists of 17 self-reported items that screen for the presence of PTSD symptoms over the last month [33]. Items are scored on a 5-point Likert scale from “not at all” to “extremely severe”. The PCL-5 scores range from 17 to 80, with higher scores indicating more severe symptoms. A cut-off score of 33 is proposed to discriminate between people with or without probable PTSD [33]. An alpha of α = 0.95 was observed in the current sample for the PCL-5.

### 2.3. Procedure

Following approval from the Federation University Ethics Committee (Approval Number: A21–124), participants who experienced wildfires in the last decade, were 18+ years old, and could read and write English, were recruited into the study. A URL link was generated using the Qualtrics survey platform and was distributed via Facebook campaigns, Instagram, Reddit, LinkedIn, online community noticeboards, local newspapers, wildfire interest group sites, and using snowball sampling methods. Participation in the survey was voluntary, with no incentives being offered. A digital plain language statement about the study was presented, and participants provided consent by selecting an “I agree” button to take part. The survey took 30 min to complete and was launched between October 2021 and March 2022.

### 2.4. Statistical Method

One hundred and eighty-nine participants took part in the survey. Participants who completed only 3–48% of the entire survey (24; 12.7%) and those who were missing 100% data on the main scales (39; 23.6%) were excluded. Missing value analysis indicated that missing data for the remaining participants were Missing Completely at Random (Little’s MCAR test, *χ*^2^ = 834.59; *df* = 845, *p* = 0.59). Therefore, participants with <10% of data missing on the dependent variables were included, and missing values were replaced by computing the series mean for missing items (ISI = 3 participants, PHQ9 = 2 participants, GAD7 = 1 participant, PCL5 = 1 participant, PSQI = 5 participants) [34].

An inspection of histograms, Probability Plots (P-P), and scatterplots indicated a normal distribution of all scales except the DDNSI, which was found to be positively skewed [35].

Descriptive statistics, including frequencies, means, and standard deviations, for each dependent variable were obtained using the IBM SPSS for Windows (Version 26). Analysis of covariance ANCOVA and post-hoc analyses were used to compare the mean differences in scores for participants from the three countries for the GAD-7, ISI, PHQ-9, PSQI, and PCL-5 scales. As indicated above, not all participants completed all scales and/or supplied all demographic variables, and the number of participants in each analysis may vary from the total number for each country (44 participants from Australia, 27 from Canada, and 55 from the USA). For example, only 87 participants from the three countries completed the DDNSI scale.

## 3. Results

### 3.1. Descriptive Statistics for Demographic Variables for Australia, Canada, and the USA

Frequencies on demographic variables for each country were calculated. Table 1 shows that more participants from the USA held a bachelor’s degree (35.2%) than participants from either Australia (27.3%) or Canada (22.2%). However, a greater percentage of participants from Australia held a postgraduate degree (25%) than participants from either Canada (7.4%) or the USA (16.7%). In addition, following the conversion of income currency from AUD to USD, a higher percentage of participants from Canada (34.6%) earned USD 26,290 to 49,290 per year than participants from either Australia (27.9%) or the USA (24.1%). Nevertheless, more participants from Australia (16.3%) earned USD 98,580 or more per year than participants from Canada (7.7%) and the USA (3.7%). With respect to employment status, a greater percentage of participants from Australia (63.6%) reported being employed than participants from both Canada (55.6%) and the USA (48.1%). Furthermore, a lower percentage of participants were found to be unemployed in Australia (2.3%) than participants from either Canada (11.1%) or the USA (13%). Recency of fires was coded as wildfires taking place less than 12 months ago or wildfires taking place more than 12 months ago [16,18]. Forty-three (97.7%) participants from Australia reported experiencing wildfires more than 12 months ago. Twenty-one (77.8%) participants from Canada reported experiencing wildfires less than 12 months ago, and 5 (18.5%) reported experiencing wildfires more than 12 months ago. Finally, 17 (30.9%) participants from the USA reported experiencing wildfires less than 12 months ago, and 38 (69.1%) reported being affected by wildfires more than 12 months ago.

### 3.2. Frequencies of Variables for Australia, Canada, and USA

The severity of symptoms and frequencies on the DDNSI, GAD-7, ISI, PHQ-9, PSQI, and PCL-5 were calculated for each country. Cut-off scores were utilized as specified in each scale in Section 2.2. No significant differences were found for nightmare symptoms between participants from Australia, Canada, and the USA (Table 2). However, a higher percentage of participants from Canada (42.1%) reported more nightmare symptoms than participants from Australia (30.8%) and the USA (21.4%). Table 2 also shows that a higher percentage of participants from the USA (47.8%) reported significantly more anxiety symptoms at the severe level than participants from Australia (24.3%) and Canada (18.5%). Similarly, a higher percentage of participants from the USA (21.8%) reported significantly more insomnia symptoms at the severe level than participants from Australia (2.3%) and Canada (11.1%). Furthermore, a significantly higher percentage of participants from the USA (26.4%) had depressive symptoms at the moderate–severe level than participants from Australia (17.55%) and Canada (11.5%). In addition, even though a higher percentage of participants from the USA (91.7%) reported having “poor sleep” than participants from Australia (80.5%) and participants from Canada (88.9%), these differences were not significant. Finally, a larger percentage of participants from the USA (88.9%) scored significantly higher at the “above the clinical threshold” for PTSD symptoms on the PCL-5 scale than did participants from Australia (48.6%) and Canada (75%) (refer to Table 2).

### 3.3. Mean Differences in Symptom Presentations between the Three Countries

Mean differences in symptom scores between participants from the three countries were examined using analyses of covariance (ANCOVA), where GAD-7, ISI, PHQ-9, PSQI, and PCL-5 were entered as dependent variables. The survey country was entered as a fixed factor in the analyses, and gender, education level, employment, income, and recency of fires were entered as covariates. Assumptions of homogeneity of variance (Levene’s test) and normality tests were both met. Table 3 shows the findings for the analyses of differences between participants from Australia, Canada, and the USA on the dependent variables after controlling for demographic variables.

The ANCOVA analysis showed that GAD-7 scores were significantly different between the three countries (*F* (2, 106) = 4.46, *p* = 0.014), and remained significant even after entering the covariates into the model (refer to Table 3). Pair-wise post-hoc comparisons conducted at *p* < 0.05 revealed no significant differences in scores on the GAD-7 between participants from Australia and Canada (*p* = 0.85) and between participants from Canada and the USA (*p* = 0.64). However, participants from the USA scored significantly higher than participants from Australia (*p* = 0.008) on the GAD-7. Only employment was found to be a significant covariate in this model (*p* = 0.015). Employment reduced the likelihood of higher scores on the GAD-7, but the country of survey better accounted for overall differences between scores.

Similarly, ANCOVA revealed a significant difference between the three countries for ISI scores (*F* (2, 120) = 6.24, *p* = 0.003), and after entering all the demographic variables as covariates, the ISI scores remained significantly different for Australia, Canada, and the USA (see Table 3). Pair-wise post-hoc comparisons showed that scores on the ISI for participants from the USA were significantly higher than scores for participants from Australia (*p* = 0.003). Participants from Canada showed no significant difference in the ISI scores from participants from Australia (*p* = 0.06) or the USA (*p* = 1.00). Income was found to be a significant covariate in this model (*p* = 0.006). Higher income reduced the likelihood of higher scores on the ISI, but main differences were better accounted for by country of survey.

ANCOVA showed that the PSQI scores differed significantly between Australia, Canada, and the USA, (*F* (2, 114) = 3.81, *p* = 0.025). However, this difference was no longer significant when the demographic variables were entered as covariates (refer to Table 3). Both gender (*p* = 0.021) and income (*p* = 0.016) were significant in the ANCOVA model, indicating that female gender and lower income accounted for more of the differences in scores than the country of survey.

In addition, scores on the PCL-5 were found to be significantly different between the three countries, (*F* (1, 102) = 4.81, *p* = 0.01), and this difference continued to be significant even with the addition of the covariates to this model (Table 3). Pair-wise post-hoc comparisons showed that scores on the PCL-5 for participants from the USA were significantly higher than scores for participants from Australia (*p* = 0.02). However, no significant differences were observed for scores on the PCL-5 between participants from Australia and Canada (*p* = 0.26) and between participants from the USA and Canada (*p* = 1.00). Gender was the only significant covariate in this model (*p* = 0.034), indicating that female gender accounted for some of the differences between countries in PCL-5 scores.

The PHQ-9 was the only dependent variable that was not significant in the ANCOVA analysis, and none of the covariates were found to be significant (Table 3).

A Kruskal–Wallis analysis was used to assess the differences between participants’ scores on the DDNSI scale across the three countries. The analysis showed that there were no significant differences between the three countries (*χ*^2^ (2, *n* = 87) = 1.06, *p* = 0.589) on the DDNSI. To assess the likelihood of affecting the outcome in the former analysis, Kruskal–Wallis analyses were used to examine associations between the DDNSI scores and all demographic variables. No significant associations were found between any of the demographic variables and the scores on the DDNSI, and therefore, the non-significant results for the comparisons between countries are unlikely to have been due to any effects of the demographic factors.

## 4. Discussion

The aim of this study was to compare the frequency and severity of anxiety, depression, insomnia, sleep quality, nightmares, and PTSD symptoms among participants affected by wildfires in Australia, Canada, and the USA. The descriptive data confirmed differences in frequencies on demographic variables between the three countries with an unequal number of participants completing the survey; more participants from the USA (*n* = 55) took part in the online survey than participants from Australia and Canada (*n* = 44, *n* = 27, respectively). Furthermore, the overall sample consisted of more females (81%) than males (18.3%).

Studies conducted in the field of wildfires and the impact they have on peoples’ mental health have been consistent in demonstrating elevated rates of multiple mental health conditions, such as depression, anxiety, sleep difficulties, and PTSD, in the aftermath of wildfires [13,15,16,17,18,21]. In comparing the three countries’ anxiety symptoms, significant differences among Australia, Canada, and the USA were found. Approximately 50% of participants from the USA reported significantly more “severe” anxiety symptoms than participants from Australia and Canada (*p* = 0.001). Similarly, a higher percentage of participants from the USA reported significantly more severe symptoms of insomnia (clinically moderate–severe level, *p* = 0.002), depression (moderately severe, *p* = 0.021), and trauma symptoms (above clinical threshold, *p* = 0.001) than their counterparts in Australia and Canada. Although the current findings are in line with previously reported studies in the field of wildfires and mental health, the reported percentages are somewhat different from those observed in the literature. This is not surprising given the various methods used in each study and the unique characteristics of each sample.

Notably, Owusu et al. [36] found that 42.5% of their sample (*N* = 186) reported symptoms of anxiety after the fires. Insomnia is one of the most prevalent mental health conditions after the fires [37], with studies reporting different percentages: 49.2% [13], 63.0% [38], and 43.6% [39]. Another mental health condition that is also repeatedly seen following the fires is depression. For example, using the PHQ-9, depression has been reported at various rates, 25.5% [39], 45.0% [40], and 32.5% [17], by several studies. Moreover, the current findings are also in line with other studies in relation to PTSD symptoms. For instance, a study by Belleville et al. [39] found that three months after the Fort McMurray fires, nearly 60% of survivors (*N* = 379) suffered from PTSD, and 29.1% (*n* = 55) met the clinical diagnostic criteria for PTSD [39]. Furthermore, different percentages of PTSD symptoms, 46.7% [38], 39.6% [40], 12.8% [41], and 77.88% [13], have been reported at different times after the fires took place.

When exploring the mean differences between Australia, Canada, and the USA, anxiety symptoms remained significantly different between the three countries, with employment being a significant contributor to this difference. In inspecting the data, the unemployment rate was higher in the USA sample. It is well documented in the literature that being unemployed leads to lower income, which plays a major role in heightening anxiety levels. For example, five years following the Fort McMurray fires, Owusu et al. [36] (*N* = 186) found that unemployed survivors were seventeen times more likely to develop anxiety symptoms (Odd’s Ratio = 16.62; 95% C.I. 1.23–223.67) in comparison to those who were employed. When communities are impacted by wildfires, they are subjected to displacement, job relocation, and/or job loss, leading to lower income [42]. This can, in turn, lead to higher levels of anxiety, not only due to the trauma of wildfires but also due to the losses associated with them [42]. Research indicates that some of the most reportedly encountered challenges by survivors after the fires include access to housing and gaining employment [42,43,44]. In the current sample, insomnia was also found to be significantly different, with income as a significant contributor to the elevated levels of both insomnia and sleep quality between the three countries. A review of studies [37] found that those who reported high levels of insomnia were also more economically disadvantaged. Long-term displacement from one’s home while their property is being rebuilt, uncertainty about employment, and loss of assets can cause major disruption to one’s sleep routine, sleep hygiene, and eventually sleep quality [13,45]. The mean scores on sleep did not differ between the three countries in the current sample. Research shows that pre-levels of sleep quality determine the level of traumatic symptoms in the aftermath of traumatic experiences [46]. The current study did not account for pre-fire levels of sleep quality; therefore, this may have contributed to the non-significant differences between Australia, Canada, and the USA.

Post-traumatic stress disorder symptoms were also found to be significantly different between the three countries in the current study. Gender was the only significant variable in this model. The association between gender and PTSD symptoms is well established, with females reporting higher rates of PTSD than males in wildfire survivors [16,38]. More specifically, one study found that the prevalence of PTSD symptoms was 12.8%, and females reported higher rates of PTSD than males (14.9%, 8.7%, respectively) [41]. Similarly, another study (*N* = 2085) also reported similar findings [47]. Evidence suggests that women are more likely to experience sexual assault, incidents of violence, and childhood trauma than men; this, in turn, can lead to the buildup of cumulative trauma, possibly exacerbating reported differences between males and females in PTSD symptoms following the trauma of wildfires [21,48,49].

Even though the current study found a significant difference in percentages in depression scores, exploring the mean scores of depression between the three countries did not show significant differences. This is perhaps a function of the small sample size in the current study compared to other studies.

Similarly, no significant differences were found for the three countries on nightmare symptoms. This contrasts with what is reported in the literature [13,15]. It is possible that nightmares change gradually from the content of events to symptoms that overlap with other mental health conditions in the weeks and months following the trauma [50,51].

Recency of fires was not found to be a significant contributor to differences between the three countries in the current study. This is contrary to what has been reported by other studies [16,18,20,21,36,38,39,40,41,42]. One line of research suggests that people “bounce back,” and mental health conditions such as PTSD wane rapidly in the first few months after disaster [52]. No complementary measures, such as coping/resilience scales, were used in the current study, which may have better explained the current findings in relation to the recency of fires [21].

Overall, participants from the USA reported significantly more severe symptoms of anxiety, insomnia, depression, and PTSD than participants from both Canada and Australia. In explaining the findings of the current study, two hypotheses may be considered. First, the number and magnitude of disastrous events that have occurred in the USA in the last two decades in comparison to Canada and Australia, and second, the disparities between the three countries in terms of the availability of resources for survivors, preferences for self-help, differences in land management and cultural practices, and preparedness levels for wildfires.

In the last twenty years, the USA—unlike Canada and Australia—experienced economic recessions and natural disasters on a national level, more so than the other two countries [53]. Findings from the 2016 World Mental Health Survey from 24 countries across the globe found that the USA (82.7%) was second only to Ukraine (84.6%) in its citizens being affected by any type of trauma [54].

Availability and accessibility to mental health resources are not paralleled nor linear in the three countries. For example, Australia provides Medicare to all its citizens, which is affordable, while the USA government provides Medicare only to people with low income and to retirees [55]. Some of the most reported challenges by survivors of California’s wildfires in 2017 and 2018 were: lack of accessibility to safe and secure rental properties, shelters, and hostels following the fires; loss of jobs; difficulties in accessing basic health needs; and delay in response from insurance companies. This led to an exacerbation of psychological symptoms and stress levels [42]. Furthermore, survivors of wildfires show a preference for self-help. For example, a Canadian study of 1510 evacuees from the 2016 Fort McMurray wildfires found that 26.8% of the sample preferred self-help to seeking help from a health professional, while 47.2% of the sample preferred self-help to receiving medications [56]. Other studies showed similar findings in that survivors of different types of traumas who experienced symptoms of depression, substance dependence, insomnia, anxiety, and PTSD reported a preference for self-help to seeking help from health professionals [57,58,59,60].

The higher rates of mental illness in the USA sample may also be related to policies associated with forest and land management. Prior to the European settlement, cultural burning was long known in the indigenous communities in the USA and Australia as part of “caring for the land” [61]. These cultural practices have been overlooked and ignored with the rise of the Industrial Revolution. In the state of California/USA, more than 129 million trees have died since 2010, as forest management has been neglected and overlooked [62]. The USA commission reported that 27 million trees have died nationally since 2016. There is a call for adopting historical and cultural practices such as planned burnings to preserve the land and reduce the magnitude of wildfires [62]. Another major discrepancy between the three countries is the different approaches they adopt in managing disasters. Experts of wildfires report on how Canada’s forests have been logged and abandoned, leaving the land more vulnerable to accumulating tons of flammable fuel for wildfires [63]. Underwood, a wildfire expert, states that academics and environmentalists adopt the emergency response or what is referred to as the “American approach”—wait for the disaster to take place, then try to contain it—while wildfire experts support the “Australian approach”, which recognizes that wildfires cannot be prevented; however, they can be mitigated through sound land management [63]. If the fire grounds are better prepared, then the consequences of wildfires will be easier to manage, safer, and cheaper to control [63].

In Australia, unlike Canada, there is an awareness about land care among vulnerable communities [64]. Preparedness for the fire season is encouraged in Australia to become not only a seasonal but also a regular practice and a way of living among farmers [65]. There is also an awareness in the Australian community about the implications of maximizing crop productivity, density of crop per hectare, planned burnings, and the impacts these practices could have on communities [65]. Australians are now working in partnership with indigenous groups to implement traditional knowledge and wisdom to care for the land [61,66]. There is a recognition that cultural burning/savannah burning has been a successful tool in land management [66]. The practice of savannah burning, whereby smaller fires are lit to suppress the occurrence of larger and out-of-control fires, is now being adopted and applied in Canada [67]. Better land management is not only about minimizing the impact of wildfires but also leads to creating job opportunities and building stronger communities [67].

Researchers are now calling for a new model called the “developmentalist model”, whereby land management is not only the responsibility of the government but also the responsibility of communities at large. It is about changing values and raising awareness about the relationship between practices and consequences pertaining to sustainable forest management [68]. The benefit–cost analysis should be applied and reflected upon when discussing the market value of forests, and a recognition of the social, cultural, and economic values of forests should all be considered before the implementation of policies [68].

Implications: clinicians treating survivors of wildfires should have sufficient training in recognizing symptoms of anxiety, depression, insomnia, sleep quality, nightmares, and PTSD. Knowledge about barriers to seeking professional help is imperative, as delays in seeking treatment may lead to the progression of symptoms and the development of chronic psychopathology [39,40,41]. Countries that showed a more severe presentation of mental health conditions, for example, the USA, may benefit from reviewing policies associated with the availability of resources and forest and land management practices.

Limitations: the current cross-cultural survey is based on retrospective data, which may have masked any pre- or post-traumatic events following the fires that could have contributed to the reported findings. Longitudinal studies are needed to understand the factors at the pre-, peri-, and post-stages following the fires, which can impact the outcomes of mental health for wildfire survivors. Another limitation of the study was the absence of measures of coping/resilience, which could have shed some light on the differences between the three countries. Finally, the differences in the timeline of fire occurrences for the three countries may have contributed to the current findings. While the USA and Canada faced wildfires in 2022–2023, Australia’s latest wildfires were in 2020.

## 5. Conclusions

Overall, the current cross-cultural sample showed differences in anxiety, insomnia, and PTSD symptoms. Variables such as gender, income, and employment contributed partially to the observed differences. The current findings also indicated that participants from the USA reported more severe levels of mental health conditions than their counterparts in Australia and Canada. The differences between Australia, Canada, and the USA may be attributable to differences in the availability of resources to survivors of wildfires and differences in policies pertaining to forest management and land practices. International collaborative research will offer one way of communication in responding to and recovering from wildfire disasters, as there are valuable lessons to be learned from Australia, Canada, and the USA.

## Figures and Tables

**Table 1 ijerph-21-00038-t001:** Frequencies of gender, education, employment, income, and recency of fires for Australia, Canada, and the United States of America.

Variables	Australia *n* (%)	Canada *n* (%)	USA *n* (%)
Gender			
Males	13 (29.5)	5 (18.5)	5 (9.1)
Females	31 (70.5)	22 (81.5)	49 (89.1)
Non-binary	----	----	1 (1.8)
Total (n)	44	27	55
Education level			
Primary school	----	----	1 (1.9)
High school	5 (11.4)	6 (22.2)	12 (22.2)
Certificate/diploma	16 (36.4)	13 (48.1)	13 (24.1)
Bachelor’s degree	12 (27.3)	6 (22.2)	19 (35.2)
Postgraduate degree	11 (25)	2 (7.4)	9 (16.7)
Total (*n*)	44	27	55
Employment			
Student	1 (2.3)	----	1 (1.9)
Employed	28 (63.6)	15 (55.6)	26 (48.1)
Unemployed	1 (2.3)	3 (11.1)	7 (13)
Looking for work	3 (6.8)	1 (3.7)	1 (1.9)
Retired	11 (25)	8 (29.6)	19 (35.2)
Total (*n*)	44	27	54
Income			
AUD 0 income	1 (2.3)	----	---
AUD 1 to 20,799 per year	4 (9.3)	3 (11.5)	14 (25.9)
AUD 20,800 to 41,599 per year	12 (27.9)	3 (11.5)	10 (18.5)
AUD 41,600 to 77,999 per year	7 (16.3)	9 (34.6)	15 (27.8)
AUD 78,000 to 155,999 per year	12 (27.9)	9 (34.6)	13 (24.1)
AUD 156,000 or more per year	7 (16.3)	2 (7.7)	2 (3.7)
Total (*n*)	44	26	54
Recency of fires			
Less than 12 months	----	21 (77.8)	17 (30.9)
More than 12 months	43 (97.7)	5 (18.5)	38 (69.1)
Total (*n*)	43	26	55

**Table 2 ijerph-21-00038-t002:** Frequencies and percentages of the DDNSI, GAD-7, ISI, PHQ-9, PSQI, and PCL-5 for Australia, Canada, and the United States of America.

Variables	Australia *n* (%)	Canada *n* (%)	USA *n* (%)	*χ*^2^ (*df*), *p*
DDNSI				
No nightmares	18 (69.2%)	11 (57.9%)	33 (78.6%)	12.23 (2), 0.002
Nightmare disorder	8 (30.8%)	8 (42.1%)	9 (21.4%)	0.08 (2), 0.961
Total (*n*)	26	19	42	
GAD-7				
Minimal anxiety	16 (43.2%)	6 (22.2%)	8 (17.4%)	5.60 (2), 0.061
Mild anxiety	8 (21.6%)	11 (40.7%)	9 (19.6%)	0.50 (2), 0.779
Moderate anxiety	4 (10.8%)	5 (18.5%)	7 (15.2%)	0.87 (2), 0.646
Severe anxiety	9 (24.3%)	5 (18.5%)	22 (47.8%)	13.17 (2), 0.001
Total (*n*)	37	27	46	
ISI				
No clinical insomnia	17 (38.6%)	3 (11.1%)	5 (9.1%)	13.76 (2), 0.001
Subthreshold insomnia	12 (27.3%)	8 (26.6%)	19 (34.5%)	4.77 (2), 0.092
Clinical moderate insomnia	14 (31.8%)	13 (48.1%)	19 (34.5%)	1.35 (2), 0.510
Clinical severe insomnia	1 (2.3%)	3 (11.1%)	12 (21.8%)	12.88 (2), 0.002
Total (*n*)	44	27	55	
PHQ-9				
Minimal depression	13 (32.5%)	3 (11.5%)	5 (9.4%)	8.00 (2), 0.02
Mild depression	8 (20%)	8 (30.8%)	17 (32.1%)	4.91 (2), 0.08
Moderate depression	8 (20%)	10 (38.5%)	8 (15.1%)	0.308 (2), 0.86
Moderately-severe depression	7 (17.5%)	3 (11.5%)	14 (26.4%)	7.75 (2), 0.021
Severe depression	4 (10%)	2 (7.7%)	9 (17%)	5.20 (2), 0.074
Total (*n*)	43	27	54	
PSQI				
Poor sleepers	33 (80.5%)	24 (88.9%)	44 (91.7%)	5.96 (2), 0.05
Good sleepers	8 (19.5%)	3 (11.1%)	4 (8.3%)	2.800 (2), 0.247
Total (*n*)	41	27	48	
PCL-5				
Below clinical threshold	18 (51.4%)	6 (25%)	5 (11.1%)	10.83 (2), 0.004
Above clinical threshold	17 (48.6%)	18 (75%)	40 (88.9%)	13.52 (2), 0.001
Total (*n*)	35	24	45	

Note. DDNSI = Disturbing Dream and Nightmare Severity Index; GAD-7 = Generalized Anxiety Disorder Questionnaire; ISI = The Insomnia Severity Index Scale; PHQ-9 = The Patient Health Questionnaire; PSQI = Pittsburgh Sleep Quality Index; PCL-5 = PTSD Checklist for DSM-5 Scale. *χ*^2^ = Chi-square; *df* = degrees of freedom; *p* = <0.05.

**Table 3 ijerph-21-00038-t003:** Means and standard deviations of the dependent variables and results of ANCOVA comparisons between participants from Australia, Canada, and the United States of America.

Dependent Variables and Covariates	Countries			*F* (*df*), *p*
	Australia *M* (*SD*)	Canada *M* (*SD*)	USA *M* (*SD*)	
GAD-7	7.55 (6.96)	8.88 (5.50)	12.52 (6.87)	
				5.00 (2, 106), 0.009
Gender				0.26 (1, 106), 0.609
Education level				0.27 (1, 106), 0.602
Employment				6.19 (1, 106), 0.015
Income				2.93 (1, 106), 0.090
Recency of fires				0.18 (1, 106), 0.671
ISI total	10.52 (6.79)	15.00 (6.12)	16.25 (6.68)	
				6.00 (2, 120), 0.003
Gender				1.71 (1, 120), 0.193
Education level				0.04 (1, 120), 0.844
Employment				0.99 (1, 120), 0.321
Income				7.90 (1, 120), 0.006
Recency of fires				0.17 (1, 120), 0.681
PHQ-9	9.02 (7.17)	10.56 (5.42)	12.58 (6.60)	
				1.71 (2, 118), 0.186
Gender				1.25 (1, 118), 0.267
Education level				0.33 (1, 118), 0.566
Employment				0.58 (1, 118), 0.448
Income				3.89 (1, 118), 0.050
Recency of fires				0.00 (1, 118), 0.961
PSQI	8.18 (4.22)	9.48 (3.31)	10.90 (4.40)	
				2.47 (2, 112), 0.890
Gender				5.46 (1, 112), 0.021
Education Level				0.02 (1, 112), 0.882
Employment				0.11 (1, 112), 0.739
Income				6.06 (1, 112), 0.016
Recency of fires				0.18 (1, 112), 0.674
PCL-5	39.53 (17.60)	46.08 (16.62)	51.82 (15.32)	
				4.01 (2, 103), 0.021
Gender				4.61 (1, 103), 0.034
Education Level				0.36 (1, 103), 0.551
Employment				3.26 (1, 103), 0.074
Income				2.81 (1, 103), 0.097
Recency of fires				1.17 (1, 103), 0.283

Note. GAD-7 = Generalized Anxiety Disorder Questionnaire; ISI = The Insomnia Severity Index Scale; PHQ-9 = The Patient Health Questionnaire; PSQI = Pittsburgh Sleep Quality Index; PCL-5 = PTSD Checklist for DSM-5 Scale; *M* = mean; *SD* = standard deviation; *F* = *F*-test; *df* = degrees of freedom; *p* = <0.05.

## Data Availability

The data presented in this study are available on request from the corresponding author. The data are not publicly available due to ethical restriction.

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
