# Peer review of "Differences in Anxiety, Insomnia, and Trauma Symptoms in Wildfire Survivors from Australia, Canada, and the United States of America"

_ijerph, 2023, doi:10.3390/ijerph21010038_

Round 1
Reviewer 1 Report
Comments and Suggestions for Authors
The present manuscript reports differences in Anxiety, Insomnia, Sleep Quality and Trauma Symptoms in Wildfire Survivors from Australia, Canada and USA. The topic of the study is very interesting; paper is well written but I have several concerns that are mentioned below:
1. Information on statistics should be moved from Results to the Methods. Further, statistics should be expanded – on lines 146-147 ANCOVA is mentioned for DDNSI analyses, while lines 228-230 state that Kruskal Wallis analysis was presented for DDNSI
2. Table 1 – why total numbers for each variable is different, and mostly not 126?
3. Table 2 – no information on DDNSI
4. Lines 228-230, if non-parametric test was used to assess the differences between participants’ scores on the DDNSI measure, how recency of fires was accounted?
5. Line 231 – what is n=87?
6. My main concern is discussion - Study is descriptive. Authors do not assess if studied parameters are influenced not only by recency of fires but demographic variables (especially, economic and employment status) that they found to differ significantly among countries. It’s not done/discussed and not even mentioned in limitation. At the same time most of discussion concerns resources availability; Indeed, resource availability is an important point to consider but it could have been assessed at least with a single question about availability of healthcare resources in a timely manner. Without such data in a studied population devoting “whole” discussion to it is misleading and assumptions are very arbitrary. Discussion should be modified.
Author Response
Thank you for taking the time to read the manuscript. Please find attached table addressing your comments.
Thank you again

Reviewer 2 Report
Comments and Suggestions for Authors
Dear authors,
First of all I woulf like to congratulate you for the good work! this paper is innovative and useful for further research as it touches an interdisciplinary issue that combines health, enrivonment and crisis.
At the end of the Introducation it would be useful to write a few lines about the research gap and the limited research activity on this field.
Moreover, at the chapter of methods it is importnat to expand the information given adding something about the method, sampling, research tools and how they are supposed to be suitable for this research, the way you analyzed data and ethics.
All the rest in this article are excellently presented and suitable for publication.
Thanks for the opportunity to read your research!
Kind Regards
Author Response

(The authors gave the same response as above.)

Reviewer 3 Report
Comments and Suggestions for Authors
Thank you for the opportunity to read this manuscript. I think it has the potentially to be an interesting paper and helpful for us to understand the impact of wildfires on mental health. However, there are a number of issues with the manuscript that must be addressed before it can be considered for publication. Below I have listed my major concerns and some minor points. I hope the author finds them constructive and useful.
Introduction.
1. In the introduction, the authors spend three paragraphs describing multiple wildfire occurrences in the three studied regions. While this information is interesting and gives a picture of the level of destruction wildfires can cause, I think it is too much. Try to reduce this to a single paragraph, perhaps describing one example wildfire from each country.
2. Connected to my first comment, the main focus of the paper is not indeed the wildfires but rather upon their effect upon the mental health of those who experienced them; however, the authors only use one short paragraph (lines 79-86) to address this. I’d like them to expand this section and give more details, perhaps drawn from the literature.
Methods
3. Section 2.2. Please list all of the demographic items collected including the ways in which they were categorized. Especially important is “recency of fires”, as this forms part of your statistical analysis.
4. Section 2.2. Please give more details about the contents and scoring of each of the instruments.
5. Please give information about how you categorized the results of each scale; for example, “Participants were categorized as Poor sleepers and Good sleepers based on PSQI scores of…”
6. Section 2.3. Please give details about how the online survey was conducted. Did you use a particular software? Website? Was there any incentivization for participation.
Results
7. At the beginning of the Results section, the authors have the statistical methods. Please consolidate the statistical methodology as a section within Methods. Please try to include information about all the statistical tests used and the software used for the analyses.
8. Section 3.1. Please put the demographic information into a table. You can then remove any demographic information from the manuscript’s text that has no bearing on the analyses included in the study. For example, you compare by country, but you don’t compare by education level, economic status, or even gender (which might be very interesting); thus, much of the demographic information given in this section isn’t really pertinent for understanding the rest of your paper.
9. Section 3.3. In the results section, it is appropriate to include the data in the text. For example: “Likewise, a higher percentage (##%) of participants from the USA reported significantly more insomnia symptoms on the severe level than participants from Australia (##%) and Canada (##%).”
10. Tables 1 and 2. In the right-hand column giving the chi-alphas and p=value, you can remove the “χ2(2) =” and “p =” “F” because you have them in the column’s header. And please define all abbreviations in the tables’ footer (df = degrees of freedom, etc.)
11. Regarding the statistical analyses, the authors performed an analysis of covariance, however they neglect to explain any way in which the data sets were tested for normality (Shapiro-Wilk), homogeneity of variances (Levene), etc., which could cast doubt on the results. Please give this information.
Discussion
12. The discussion section lacks logical flow. This is hindered and confused by the subheadings. Moreover, the underlying message seems to be “The USA is bad. Australia is great”. The authors’ bias is too obvious (even if it is justified).
13. The authors state that: “The findings showed that participants from the USA scored significantly higher on all variables, except nightmare and depression, than participants from Australia and Canada after controlling for recency of fires.” However, no discussion of depression is given in the Discussion section.This is a major oversight.
14. Section 4.3 Magnitude of trauma. Here the authors offer tangential information on economic and social events in the US that have had a negative impact on its citizens’ mental health as a possible explanation for the findings of the study. This information doesn't really fit the logical flow of ideas; the author goes from talking about wildfires to suddenly talking about terrorism and war. I would suggest remove this section and surmise it into 1 or 2 sentences as part of a larger discussion into possible explanations for the findings.
15. The major thing missing from this paper is an honest discussion of the role of bushfires in Australia. They form, according to the Australian climate service “a natural, essential and complex part of the Australian environment and have been for thousands of years.” https://www.acs.gov.au/pages/bushfiresMost Australians are not only aware of the dangers of bushfires, but also their necessity. Many Australians, my family members included, chose to live in homes built in areas susceptible to bushfires, with home insurance to cover this danger. Australia also has resources and education on bush fires. This must contribute somewhat to ameliorating the trauma associated with wildfires. Furthermore, bushfires in Australia are also (perhaps uniquely) part of the cultural identity among the indigenous population. As for the US, the wildfire problem seems to be related to forest mismanagement particularly in California (https://lhc.ca.gov/report/fire-mountain-rethinking-forest-management-sierra-nevada) and ignorance (the 2020 El Dorado wildfire being an example). Comparing the US and Australia, it seems like Australia is more prepared for and aware wildfires in every sense than the U.S. I think discussion along these lines would be more valuable than the justification given in 4.3, for example, as it stays within the topic of wildfires.
16. Section 4.4 Is this a really long section or should it be divided into further subheadings?
17. You state that “Findings from this study showed that wildfire survivors from Australia are in a more favorable position than survivors from Canada and the USA…” What do you mean by “favorable position”? Which findings shows this?
18. Section 5 and 6. For this journal, to the best of my knowledge the limitations are usually described within the discussion section. I think the Implications section should be part of the discussion and/or conclusion. They are not usually separate sections. Please consult the journal’s Instructions for Authors https://www.mdpi.com/journal/ijerph/instructions.
19. A possible limitation not mentioned is the high percentage of women in the study. Does this skew the results? Some studies imply that women are more susceptible to mental health problems (including depression and anxiety) than men. Do you have enough data to test this question? It might be interesting.
20. In the implications section, the language should be tempered somewhat.
Comments on the Quality of English LanguageLanguage style:
1. The manuscript needs thoroughly editing for English grammar and punctuation. Many of the problems could have been resolved by using grammar checking software. However, I also advise asking one of the coauthors/supervisors to check it for you. They should be able to help not only with the grammar and style, but also the logical flow of the paper.
2. There is a lot of repetition in the manuscript, for example: “In the present study a Cronbach’s alpha of …. was observed” “Post-hoc comparisons…” etc.
3. There are many occasions of redundancy: For example, here we have repetition of English “Despite the findings from the current study, research shows that the quality of care provided in English speaking countries such as Australia, the UK, Canada and the USA does not meet the minimal standards of clinical care, and it lacks robustness in meeting the needs of citizens in English speaking countries [26].”
4. You need to be aware of word order. For example, in Line 18-19: “following wildfires in people”
5. Some words/phrases that need to be hyphenated; e.g., “help seeking” to “help-seeking” “English-speaking countries”
6. I think it is good to use the Oxford (or serial) comma consistently throughout the manuscript.
7. In the title and elsewhere, “the” should come before “United States”
8. Be aware that a spell-check might not find a spelling mistake if you have the wrong word. For example, in line 373: “my benefit”
Author Response

(The authors gave the same response as above.)

Round 2
Reviewer 3 Report
Comments and Suggestions for Authors
I commend the authors for their excellent work in addressing my comments so thoroughly. You did an excellent job revising the manuscript. I really think it's something you can be proud of. The discussion is really useful and interesting as it gives a very wholistic and positive direction for the future. I really like what you've done. Good job!